# Antecedents and outcomes of work-family conflict: A mega-meta path analysis

**Brian K. Miller**[1]*, **Maggie Wan**[1], **Dawn Carlson**[2], **K. Michele Kacmar**[3], **Merideth Thompson**[4]

1 Department of Management, Texas State University, San Marcos, Texas, United States of America, 2 Department of Management, Baylor University, Waco, Texas, United States of America, 3 Department of Management, University of South Alabama, Mobile, Alabama, United States of America, 4 Department of Management, Utah State University, Logan, Utah, United States of America

* bm25@txstate.edu

**Data Availability Statement:** All data can be found inside the paper in Tables 1 and 2.

**Funding:** The authors received no specific funding for this work.

## Abstract

This study examines the mediating role of work-to-family conflict and family-to-work conflict between the Big Five personality traits and mental health thereby enhancing theoretical development based upon empirical evidence. Integrating Conservation of Resources theory with the self-medication hypothesis, we conducted a mega-meta analytic path analysis examining the relationships among employees' Big Five traits, work-to-family conflict and family-to-work conflict, anxiety and depression, and substance use. We produced a ten-by-ten synthetic correlation matrix from existing meta-analytic bivariate relationships to test our sequential mediation model. Results from our path analysis model showed that agreeableness and conscientiousness predicted substance use via mediated paths through both work-to-family conflict and family-to-work conflict and sequentially through depression as well as through family-to-work conflict followed by anxiety. Extroversion and openness-to-experience had relatively weaker influences on substance use through work-to-family conflict, anxiety, and depression. Neuroticism was the strongest driver of the two forms of conflict, the two mental health conditions, and substance use. From this model it can be inferred that work-to-family conflict and family-to-work conflict may be generative mechanisms by which the impact of personality is transmitted to mental health outcomes and then to substance use when analyzed via a Conservation of Resources theory lens.

## Introduction

The ever-growing body of research around work-to-family conflict began with the theoretical work [1] and has been followed by immense interest in its measurement, correlates, antecedents, and outcomes. Work-to-family conflict is a form of inter-role conflict where "participation in the work [family] role is made more difficult by virtue of participation in the family [work] role" [1] [p. 77]. Because employees face pressure from work and family simultaneously, demands in one role interfere with meeting the requirements in the other role thus resulting in conflict [2, 3]. The definition implies a bi-directional feature of employees' work

**Competing interests:** The authors have declared that no competing interests exist.

and family lives, such that in some cases there is work-to-family conflict and in other cases it is family-to-work conflict [4]. This conflict is likely to have serious consequences for one's roles in their family and at work.

With the development of valid instruments [2, 5–7] research on the antecedents, correlates, and outcomes of work-to-family conflict (WFC) and family-to-work conflict (FWC) has grown in number and type. The important directionally different flow of conflict between these two major domains of life [8] results in WFC and FWC being only moderately positively correlated at .41 [9]. It is notable that work-family conflict takes unique bi-directional forms of WFC and FWC. The former is defined as when the work demands interfere with performing family responsibilities, while the latter is defined as when family demands interfere with the performance of work duties [4]. Examples of the former include work demands like forced overtime and dealing with rude customers. Examples of the latter include family demands like caring for sick children and marital strife. These two directions of conflict have enough unique variance to act as separate and stand-alone constructs because they only overlap by about 16% [9]. A large body of literature [2, 5, 9] has documented that WFC and FWC are theoretically distinct, are predicted by different factor, and result in different consequences [2, 5, 9]. Because these two forms of conflict often relate differently to other variables [10, 11] such that they are not interchangeable.

The purpose of the current study is to link some antecedents of WFC and FWC with some outcomes of the two in a mediation model not previously examined. Although some primary studies have assessed the mediating role of WFC and/or FWC in some of these relationships, most focus on antecedents or on outcomes but not usually both. By our count there are at least 54 different antecedents and outcomes of WFC and/or FWC in published primary studies, many of which have been analyzed numerous times. This accumulation of the study of some of these relationships has led to meta-analyses of the relationships between WFC/FWC and variables such as personality traits [9, 12–16], workplace policies [15, 17, 18], issues of home life [10, 18], demographics [19, 20], various forms of stress and strain [14, 21], and psychological disorders and issues [22]. However, not every relationship examined in a WFC/FWC-based primary study has been meta-analyzed.

We assembled available meta-analytic effect sizes as a synthetic input correlation matrix to a path model that tests the mediating role of WFC and FWC in the relationship between personality and mental health tied together under one major theoretical framework. A synthetic matrix of all previously analyzed associations with WFC and FWC would require a 56-by-56 matrix consisting of 1540 unique off-diagonal meta-analytic effect sizes. The overwhelming majority of the cells of such a matrix have not been meta-analyzed. We integrate some of the loosely connected segments of the nomological network of WFC/FWC via our secondary use of meta-analytic data [SUMAD] [23] as we seek to connect some elements of the left hand side [i.e. antecedents] with other elements of the right hand side [i.e. outcomes] of the nomological network surrounding WFC/FWC. The advantage of our use of SUMAD in a path model is that it allows us to simultaneously examine numerous complex relationships based upon precise meta-analytic estimates of population parameters [23].

We make several contributions to the study of WFC using the theoretical framework of conservation of resources [COR] theory [24] supplemented by the self-medication hypothesis [Khantzian, 1987]. First, our model allows us to integrate the sometimes-disconnected fields of management, personality, clinical psychology, and public health based upon existing meta-analyses. Second, by applying COR theory [24], our study brings insight into WFC/FWC as a generative mechanism by which personality affects mental health. Third, by integrating COR theory [24] with the self-medication hypothesis [25] we empirically examine theoretical arguments in a nuanced, in-depth explanation of the mechanism underlying these relationships.

Our study addresses this issue both theoretically and empirically by examining the sequential [serial] mediating effects of both WFC and FWC as well as two mental health disorders on the personality-substance use relationship. By providing scholars with a more comprehensive understanding of the proposed sequential mediating effects of WFC and FWC, we expand extant research and connect related but not often well-integrated areas of research. To be clear, there are likely a host of other mediators of the personality-to-mental-health relationship but the focus of this study is on the role of WFC/FWC.

## Theoretical framework

Conservation of Resources [COR] theory [24, 26], argues that individuals have a natural tendency to protect their valuable resources because their personal outcomes will be impaired if resource losses occur. There are four resources in COR theory: object resources, conditions, personal characteristics, and energy [24]. Personality traits are examples of personal characteristics that can aid stress resistance [24, 27] and serve as key resources [24, 27–29] in personal struggles. Specifically, self-discipline which is embodied in the personality trait of conscientiousness is a resource [27]. For example, a person high in conscientiousness may retreat and dig more diligently into their work duties as a response to distress and conflict at work. Compared with resources that are easily depleted such as energy and time, personality traits are examples of key resources that are theorized as relatively stable resources that "provide an explanation for why some people are better than others in coping with stressful circumstances" [p.550] [30]. Thus, certain personality traits can help one to cope effectively with challenging situations [31] such as incompatible work and family demands.

Personality plays a role in the allocation of one's self to one's problems and as such is likely to be relied upon as an antidote of sorts by persons experiencing conflict in the workplace. Because one's resources are finite, individuals involved in multiple role requirements tend to experience substantial resource drain [1, 32, 33]. It is also well-known that personality predicts substance use [34, 35] and the Big Five personality traits in particular [36] have a well-validated impact on substance use [37–39]. Unfortunately, the current literature presents a complex picture of these relationships. For example, neuroticism is consistently documented to be positively associated with substance use, but the other Big Five traits have been much less consistent in the prediction of substance use [38, 40–44]. These inconsistent findings suggest the need for stronger theoretical links between the Big Five traits and substance use. In line with COR theory, we consider the Big Five traits as critical characteristics that impact employees' WFC/FWC, which will further affect anxiety, depression, and substance use.

## Nomological network

### Left side of the nomological network

The Big Five personality traits have been shown to influence both directions of conflict [12, 45, 46] and some traits play a preventive role in WFC [47]. Conscientious individuals tend to be determined, diligent, organized, and achievement-oriented [36]. The orderliness, self-discipline, and effectiveness of conscientiousness help employees effectively manage their time and simultaneous demands that arise between work and family domains, thereby triggering less WFC [46, 48–50]. Agreeable individuals are cooperative, trusting, and soft-hearted [36] and are more likely to establish interpersonal bonds with colleagues and family members. These social connections help in handling work or family struggles and thus protect employees from possible conflict in either domain [12, 51]. Openness-to-experience entails being creative, flexible, and open to new perspectives [52] and is likely to assist in problem solving or coping strategies for those who are facing simultaneous demands in both work and family roles,

resulting in lower WFC. Extroverted individuals are talkative, sociable, and highly active [53]. Individuals with high extroversion normally perceive events positively [54] and are inclined to proactively look for specific solutions to challenges such as WFC [55, 56]. Individuals with high agreeableness, conscientiousness, extroversion, and openness-to-experience deal better with the two-way demands of the work and family domains [46, 55]. These four global traits are consistently negatively related to WFC.

However, neuroticism has an opposite effect on WFC and FWC compared to the other four traits. Neuroticism consists of the tendency to have low self-esteem, irrational perfectionist beliefs, and inferior emotional adjustments such as pessimismand being temperamental [54, 57]. Individuals with high neuroticism are vulnerable to experiencing emotional turmoil and encounter negative life experiences [58]. Moreover, neurotic employees are more likely to feel worried or focus on negativity [59], leaving them with less of the calm and resilience that are necessary to accomplish their mutual work and family demands 59. Because this Big Five trait plays a role in maladaptive thoughts and behavior [60], neuroticism is positively related to WFC/FWC.

### Right side of the nomological network

Work-family conflict has been found to affect anxiety and depression [47, 61–65]. Such conflict results in "a negative state of being" (p. 355) [32], often manifesting itself in these two psychological disorders [14, 63, 66, 67]. Anxiety in this study refers to clinically diagnosed generalized anxiety disorder (GAD), often characterized by excessive fear or worry that interferes with one's daily activities [68]. Depression in this study refers to clinically diagnosed major depressive disorder (MDD), a common medical illness with the symptoms of profound sadness and loss of interest that negatively affect how one feels, thinks, and acts [68]. There is a well-documented body of research on the relationship between WFC/FWC and both anxiety and depression [14, 47, 63, 64, 69–72] supporting the notion that that both WFC and FWC are positively related to anxiety and depression.

Conflict at work or home also positively relates to employees' substance use [66, 70, 73–76]. The United State loses more than $400 billion annually in low work productivity, absenteeism, increased health care expenses, law enforcement, and criminal justice costs [77] because of the use of psychoactive substances such as alcohol and illicit drugs [e.g., steroids, cannabis, stimulants, opioids, sedatives, and hallucinogens]. There were 19.7 million American adults engaged in substance use in 2017 [78]. The prevalence of substance use in the United States is both detrimental and costly to organizations [79, 80] and therefore the importance of understanding factors that predict substance use cannot be understated. In this study, substance use refers to an excessive use of alcohol and/or a sub-clinical level of use of illicit drugs and/or abuse of prescription medication, none of which could be considered severe enough to rise to the level of a substance use disorder but all of which are likely to cause problems with one's family life or work or both. We suggest that the self-medication hypothesis [25, 81], which proposes that individuals sometimes attempt to alleviate the anxious or depressive symptomatology through the use of illicit substances [82, 83], can help integrate these three outcomes of WFC/FWC still under the heading of COR. In sum, when individuals are at a resource loss state [depression and anxiety] they look to other resources (substances) to fill that void as a considerable number of studies have generally supported a positive relationship between anxiety, depression, and substance use [84]. For purposes of simplicity we hereafter refer to anxiety, depression, and substance use simply as mental health outcomes except when referring to specific hypotheses or model linkages or to properly differentiate them from each other or from other aspects of mental health.

## Simple mediation

In our overall test of the model, we focus on WFC and FWC as simple mediators of the relationships between personality with anxiety and depression as well as a sequential mediator in the relationship between personality, WFC/FWC, anxiety/depression, followed by substance use. Consistent with COR theory, the interference between work and family reflects resource depletion, as the time, strain, and behavioral interferences from one role drain the resources needed to complete duties in the other role [2, 24, 26, 32]. Negative strain can take the form of anxiety and depression [64, 69, 75]. In particular, the personality traits of agreeableness, conscientiousness, extroversion, and openness-to-experience help employees effectively utilize their other resources [24, 27, 30] to manage the interference of WFC/FWC [12, 45, 59]. The reduced conflict further decreases the occurrence of psychological disorders such as anxiety and depression [62, 63]. In contrast, employees with high neuroticism experience higher conflict, due to the fact that they are low in critical personal resources such as self-esteem and calmness that are needed to handle stressful role demands in work and family domains. Consequently, the higher level of WFC/FWC impairs employees' well-being [24] by yielding higher levels of anxiety and depression. Thus, work-to-family and family-to-work conflict are generative mechanisms by which the influence of personality is linked to anxiety and depression. These arguments lead to our first set of hypotheses:

*Hypotheses 1a-d*: *Work-to-family conflict mediates the negative relationship between the Big Five traits of (a) agreeableness, (b) conscientiousness, (c) extroversion, and (d) openness-to-experience and anxiety.*

*Hypothesis 1e*: *Work-to-family conflict mediates the positive relationship between the Big Five trait of (e) neuroticism and anxiety.*

*Hypotheses 2a-d*: *Family-to-work conflict mediates the negative relationship between the Big Five traits of (a) agreeableness, (b) conscientiousness, (c) extroversion, and (d) openness-to-experience and anxiety.*

*Hypothesis 2e*: *Family-to-work conflict mediates the positive relationship between the Big Five trait of (e) neuroticism and anxiety.*

*Hypotheses 3a-d*: *Work-to-family conflict mediates the negative relationship between the Big Five traits of (a) agreeableness, (b) conscientiousness, (c) extroversion, (d) openness-to-experience and depression.*

*Hypothesis 3e*: *Work-to-family conflict mediates the positive relationship between the Big Five trait of (e) neuroticism and depression.*

*Hypotheses 4a-d*: *Family-to-work conflict mediates the negative relationship between the Big Five traits of (a) agreeableness, (b) conscientiousness, (c) extroversion, and (d) openness-to-experience and depression.*

*Hypothesis 4e*: *Family-to-work conflict mediates the positive relationship between the Big Five trait of (e) neuroticism and depression.*

The self-medication hypothesis [25, 81] suggests that persons experiencing high anxiety or depression tend to engage in substance use, which serves a self-medicating role for reducing psychological disorders [84–86]. Building on COR theory [24, 26] and the self-medication hypothesis [25], we contend that the psychological disorders of anxiety and depression mediate the relationship between WFC/FWC and substance use. That said, the inter-role conflict between work and family domains reflects resource depletion, with the resource drain in one

domain leaving fewer resources needed to fulfill the requirements in the other domain [32]. The resource loss may generate strains in terms of higher levels of anxiety and depression to employees [14, 65, 70, 75]. In turn, anxious or depressive employees are more likely to self-medicate with substances to reduce the symptoms of anxiety and depression [84]. At least two studies have previously suggested the sequential linkage from WFC to psychological disorders to substance use. For example, in a conceptual model regarding WFC and health outcomes, it was proposed that negative emotions (prevalent to both anxiety and depression) mediated the relationship between two directions of WFC and unhealthy behaviors such as substance use [47]. Work-to-family conflict was found to be positively associated with overall emotional distress, which further resulted in increased weekly alcohol consumption [76]. We suggest that anxiety and depression act as generative mechanisms by which the impact of work-to-family and family-to-work conflict are transmitted to substance use. Therefore, we next propose that:

*Hypothesis 5a-b*: *(a) Anxiety and (b) depression simultaneously mediate the positive relationship between work-to-family conflict and substance use.*

*Hypothesis 6a-b*: *(a) Anxiety and (b) depression simultaneously mediate the positive relationship between family-to-work conflict and substance use.*

## Sequential mediation

Substance use can be partially explained by the sequential mediating effects of personality, WFC, FWC, and the psychological disorders of anxiety and depression. Guided by COR theory [24, 26, 27, 87], we expect that individuals with high levels of the personality traits of agreeableness, conscientiousness, extroversion, and openness-to-experience will be able to effectively address the simultaneous pressures from work and family domains that are mutually incompatible [88], leading to a lower level of WFC and FWC [46]. In turn, employees with lower WFC and FWC are less likely to suffer anxiety or depression [63], which reduces the occurrence of substance use [81]. Conservation of resources theory [24, 26] and the self-medication hypothesis [25] help explain the indirect effect between neuroticism and substance use. Neuroticism consists of characteristics such as insecurity, tension, and emotional instability that lead employees to be internally worried and made vulnerable by the stress of incompatible work and family demands [89]. The inefficiency of resource usage for coping with work and family challenges leads to a high level of WFC, which further increases the levels of anxiety or depression, as the stress of WFC gives rise to negative mental health consequences [75]. The sequential psychological resource loss, represented by anxiety or depression, may ultimately increase substance use [84, 90] as employees tend to adopt illicit substance use to alleviate their psychological disorders [81]. In this sequence of effects, WFC/FWC, anxiety/depression sequentially mediate the relationship between personality and substance use. Lastly, we propose the following hypotheses:

*Hypotheses 7a-d*: *Work-to-family conflict followed by anxiety sequentially mediate the negative relationship between the Big Five traits of (a) agreeableness, (b) conscientiousness, (c) extroversion, (d) openness-to-experience and substance use.*

*Hypothesis 7e*: *Work-to-family conflict followed by anxiety sequentially mediate the positive relationship between the Big Five trait of (e) neuroticism and substance use.*

*Hypotheses 8a-d*: *Family-to-work conflict followed by anxiety sequentially mediate the negative relationship between the Big Five traits of (a) agreeableness, (b) conscientiousness, (c) extroversion, (d) openness-to-experience and substance use.*

*Hypothesis 8e*: Family-to-work conflict followed by anxiety sequentially mediate the positive relationship between the Big Five trait of (e) neuroticism and substance use.

*Hypotheses 9a-d*: Work-to-family conflict followed by depression sequentially mediate the negative relationship between the Big Five traits of (a) agreeableness, (b) conscientiousness, (c) extroversion, (d) openness-to-experience and substance use.

*Hypothesis 9e*: Work-to-family conflict followed by depression sequentially mediate the positive relationship between the Big Five trait of (e) neuroticism and substance use.

*Hypotheses 10a-d*: Family-to-work conflict followed by depression sequentially mediate the negative relationship between the Big Five traits of (a) agreeableness, (b) conscientiousness, (c) extroversion, (d) openness-to-experience and substance use.

*Hypothesis 10e*: Family-to-work conflict followed by depression sequentially mediate the positive relationship between the Big Five trait of (e) neuroticism and substance use.

## Methods

### Analytic technique

Typically, primary meta-analyses are on a few bivariate effects size at a time [91] although the development of meta-analytic structural equation modeling (MASEM) [92, 93] has greatly aided in the understanding of how a cascade of variables relate to each other. Despite many primary studies on WFC/FWC using many different theoretical frameworks to explain the same relationship and certainly to explain different relationships even within the same primary study, we focus our analysis only on those model relationships that can be unified under COR theory as supplemented by the self-medication hypothesis and, of course, on relationships which are actually available as published meta-analyses. We refer to our approach as a mega-meta path analysis to differentiate it from a second-order or meta-meta-analysis that synthesizes two or more existing meta-analyses of the same bivariate relationship [94, 95]. Ours is one of the first to rely exclusively on other researchers' meta-analytic correlations rather than some primary study correlations of our own or of others combined with some meta-analytic correlations of our own or of others that are assembled into one complete input matrix suitable for analysis. To our knowledge, only one study [96] has assembled a synthetic correlation matrix solely from prior published meta-analyses to compare the predictive validity of different matrices on the same model.

Other researchers have assembled partial input correlation matrices from existing meta-analyses to supplement their own primary study effects or their own meta-analyses of other variable relationships. Several studies [17, 97–99] have conducted meta-analyses for some cells in a correlation matrix and then supplemented missing cells not meta-analyzed in their own study with previously published meta-analytic correlations from other studies to test a path model. Others [15] filled their missing cells in their own primary single-sample study input matrix with meta-analytically derived correlations from other studies to test their model. The major advantage of our mega-meta analytic path analysis is that it allowed us to test a theoretical model not previously tested in any primary study [100]. Minor advantages include the fact that the bivariate components of the input matrix are: (1) based upon very large collective sample sizes, (2) based upon a large number of primary published studies, and (3) are sample size weighted and corrected for unreliability.

This required the construction of a 10-by-10 synthetic correlation matrix [96] with 45 unique off-diagonal meta-analytically derived correlations to be used as input to a structural equation modeling test. In some cells of the input matrix, more than one meta-analysis had

been conducted on the same bivariate relationship which both facilitated and complicated the secondary use of meta-analytic data (SUMAD) [23]. The taxonomy of SUMAD [23] suggests that our use is a variation of the Type 4 variety which is a full-information MASEM (FIMA-SEM) [95]. The Type 4 variety addresses effect size heterogeneity or the variability in true score correlations in a population [101] by using bootstrapped samples built off of the cell correlation values and the standard deviation of those meta-analytic correlations to produce 80% credibility intervals for the paths and fit indices. However, it is implied [23] that both Type 3 which is a traditional MASEM [93] and Type 4 assume that only one meta-analytic correlation is available for each cell of the input correlation matrix. In most cases we have more than one meta-analysis per cell. However, in a simulation study [102] the FIMASEM technique [95] "...showed that bootstrap credibility intervals (CVs) work reasonably well, whereas test statistics and goodness of fit indices do not" thereby making the comparison of alternative models difficult. The focus of our study is on model fit as we seek to theoretically integrate rarely connected fields of research using prior validated bivariate relationships. Our study addresses effect size heterogeneity by capitalizing on the aforementioned complication of more than one meta-analytic value being available for most of the synthetic input matrix cells and therefore using MASEM on multiple permutations of our input matrices. The model tests are not nested within other models but rather the different meta-analytic input matrices allow us to test the same model with a variety of heterogenous meta-analytic correlations. The availability of different input matrices serves as a variation of the FIMASEM technique [95] such that we can compare input matrices that are known to be different rather than different models using the same singular input matrix for which the technique was designed [95] and which requires the standard deviations of the correlation cell values to produce bootstrapped results.

## Literature search

To collect meta-analytically derived correlations, we searched three online databases (PsychInfo, ABI-Inform, and Web of Science) for meta-analyses of each of the necessary 45 bivariate relationships. We used the search term strings of "meta-analy* OR quantit* rev*" combined with variations of keywords like "work-family conflict," "work-family conflict," "WFC," "family-work conflict," "family work conflict," "FWC," "Big Five," "Five Factor model," "FFM," "conscientiou*," agreeabl*," "neuroti*," "emotional stab*," "openness-to-experience," "openness," "extraver*," "extrover*," "introver*," "anxiety," "anxious*," "depres*," "substance use," and "substance abuse." The search effort uncovered 5859 unique possible meta-analyses. After the removal of 1039 duplicates from the three databases there were 5253 articles that remained eligible for screening.

## Article inclusion criteria

To be included in our study a meta-analysis had to meet four criteria: (1) it must have been written in English, (2) it had to be published in a peer reviewed scholarly journal, (3) it had to be a meta-analysis and not a primary study, and (4) it had to have reported an effect size for at least one of the 45 unique relationships necessary for our correlation matrix input. Three meta-analyses used the same unpublished dissertation [105] for their meta-analytic intercorrelations between the Big Five traits [109–111]. After coding them and comparing them to the dissertation [105] it was determined that all three correctly reported those correlations. In order to avoid duplication, the dissertation [105] was added to the coded studies list and the three published articles were removed. After all screening was completed, 12 meta-analyses were coded to fill 45 cells with 75 coded effect sizes. We used a PRISMA flow chart [112] to guide our collection of studies to code (Fig 1).

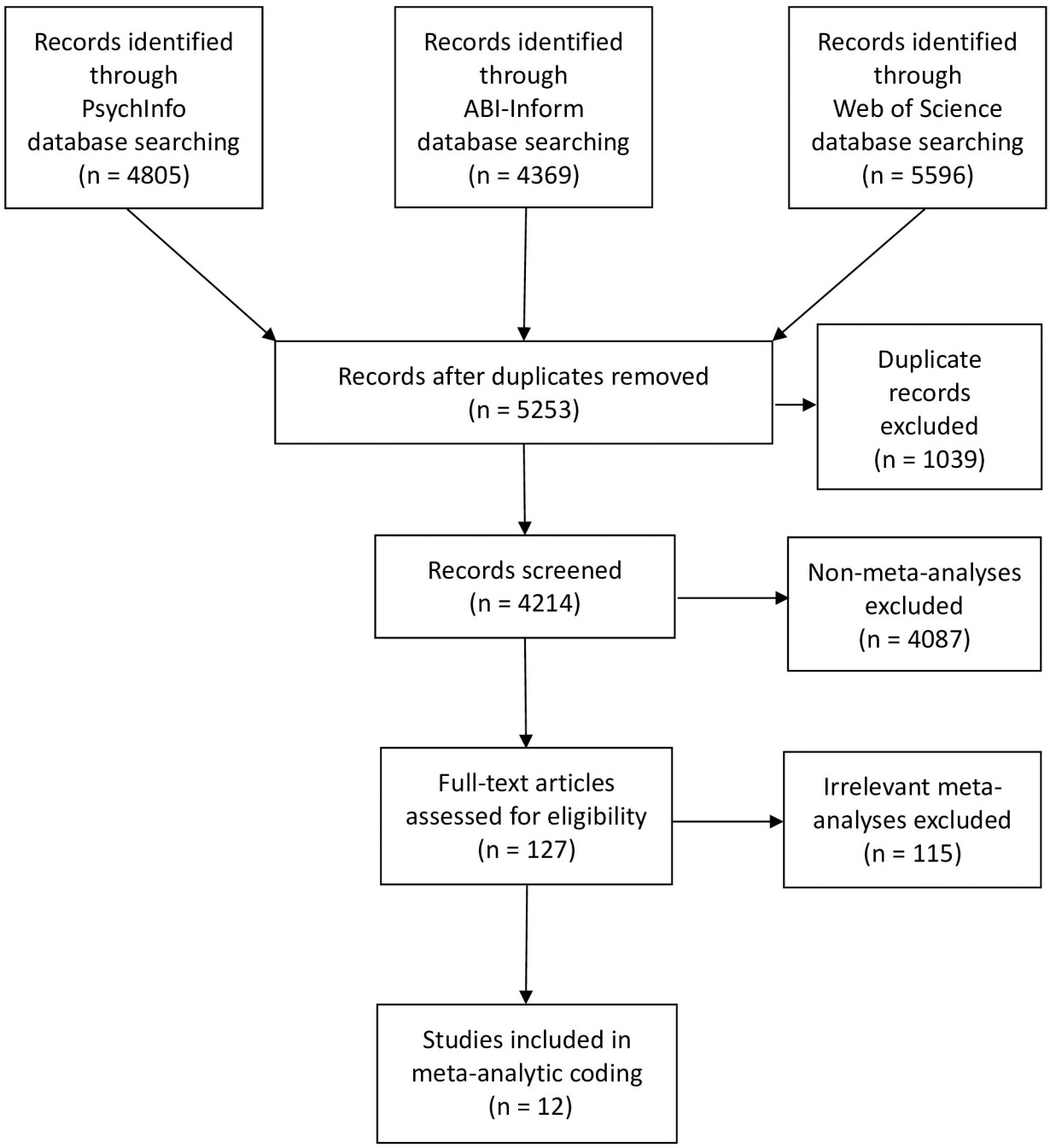

**Fig 1. PRISMA flowchart for study selection.** The steps below were used to identify and select studies for coding.

## Meta-analytic coding

The meta-analytic data extracted for analysis included: (1) type of corrected effect size (e.g. $\rho$, $r$, $d$, odds ratio [OR]), (2) corrected bivariate effect size, (3) total sample size reported [i.e. $n$] in the meta-analysis, and (4) total number of independent samples reported (i.e. $k$) in the meta-analysis. Eleven of the 45 cells were cross-coded by two of the authors. After a number of coding trials completed side-by-side, interrater agreement was perfect (See Table 1 for the coding results). The remaining 34 cells were then evenly split between the authors.

**Table 1. Effect size coding from published meta-analyses.**

| Meta-analytic relationship | Study | d | OR | r | k | n |
|---|---|---|---|---|---|---|
| WFC and FWC | Shockley and Singla (2011) [9] | – | – | .41 | 97 | 41429 |
| WFC and agreeableness | Allen et al. (2012) [12] | – | – | -.17 | 12 | 4514 |
| | Michel et al. (2011) [13] | – | – | -.18 | 13 | 5309 |
| WFC and conscientiousness | Allen et al. (2012) [12] | – | – | -.16 | 21 | 6427 |
| | Michel et al. (2011) [13] | – | – | -.22 | 20 | 6924 |
| WFC and extroversion | Allen et al. (2012) [12] | – | – | -.09 | 14 | 5112 |
| | Michel et al. (2011) [13] | – | – | -.11 | 17 | 8094 |
| WFC and neuroticism | Allen et al. (2012) [12] | – | – | .31 | 27 | 9085 |
| | Michel et al. (2011) [13] | – | – | .36 | 29 | 11775 |
| WFC and openness-to-experience | Allen et al. (2012) [12] | – | – | -.02 | 9 | 4026 |
| | Michel et al. (2011) [13] | – | – | -.05 | 11 | 4810 |
| WFC and depression | Amstad et al. (2011) [14] | – | – | .23 | 14 | 9869 |
| WFC and anxiety | Amstad et al. (2011) [14] | – | – | .14 | 3 | 4804 |
| WFC and substance use | Amstad et al. (2011) [14] | – | – | .08 | 3 | 4900 |
| FWC and agreeableness | Allen et al. (2012) [12] | – | – | -.19 | 9 | 3901 |
| FWC and conscientiousness | Allen et al. (2012) [12] | – | – | -.20 | 14 | 4494 |
| FWC and extroversion | Allen et al. (2012) [12] | – | – | -.07 | 13 | 4849 |
| FWC and neuroticism | Allen et al. (2012) [12] | – | – | .27 | 20 | 6566 |
| FWC and openness-to-experience | Allen et al. (2012) [12] | – | – | -.05 | 9 | 4026 |
| FWC and depression | Amstad et al. (2011) [14] | – | – | .22 | 10 | 6712 |
| FWC and anxiety | Amstad et al. (2011) [14] | – | – | .19 | 3 | 4804 |
| FWC and substance use | Amstad et al. (2011) [14] | – | – | .10 | 2 | 4686 |
| Agreeableness and conscientiousness | Lipnevich et al. (2017) [103] | – | – | .32 | 28 | 10113 |
| | Steel, Schmidt, & Shultz (2008) [104] | – | – | .18 | 27 | 9558 |
| | Ones (1993) [105] | – | – | .27 | 344 | 162975 |
| Agreeableness and extroversion | Lipnevich et al. (2017) [103] | – | – | .20 | 27 | 9821 |
| | Steel et al. (2008) [104] | – | – | .19 | 28 | 9912 |
| | Ones (1993) [105] | – | – | .17 | 234 | 135529 |
| Agreeableness and neuroticism | Lipnevich et al. (2017) [103] | – | – | -.11 | 28 | 10114 |
| | Steel et al. (2008) [104] | – | – | -.23 | 34 | 12036 |
| | Ones (1993) [105] | – | – | -.25 | 561 | 415679 |
| Agreeableness and openness-to-experience | Lipnevich et al. (2017) [103] | – | – | .21 | 27 | 9819 |
| | Steel et al. (2008) [104] | – | – | .10 | 28 | 9912 |
| | Ones (1993) [105] | – | – | .11 | 236 | 144205 |
| Agreeableness and depression | Kotov et al. (2010) [38] | – | – | -.06 | 25 | 21082 |
| Agreeableness and anxiety | Kotov et al. (2010) [38] | .18 | – | .05[a] | 3 | 6642 |
| Agreeableness and substance use | Hakulinen, et al. (2015) | – | 1.09 | .03[a] | 8 | 24454 |
| | Kotov et al. (2010) [38] | -.60 | – | -.27 | 25 | 23811 |
| Conscientiousness and extroversion | Lipnevich et al. (2017) [103] | – | – | .21 | 27 | 9818 |
| | Steel et al. (2008) [104] | – | – | .28 | 31 | 10974 |
| | Ones (1993) [105] | – | – | .00 | 632 | 683001 |
| Conscientiousness and neuroticism | Lipnevich et al. (2017) [103] | – | – | -.17 | 28 | 10111 |
| | Steel et al. (2008) [104] | – | – | -.33 | 37 | 13098 |
| | Ones (1993) [105] | – | – | -.26 | 587 | 490296 |
| Conscientiousness and openness | Lipnevich et al. (2017) [103] | – | – | .16 | 27 | 9816 |
| | Steel et al. (2008) [104] | – | – | .01 | 28 | 9912 |
| | Ones (1993) [105] | – | – | -.06 | 338 | 356680 |

(*Continued*)

**Table 1.** (Continued)

| Meta-analytic relationship | Study | d | OR | r | k | n |
|---|---|---|---|---|---|---|
| Conscientiousness and depression | Kotov et al. (2010) [38] | – | – | -.36 | 25 | 20747 |
| Conscientiousness and anxiety | Kotov et al. (2010) [38] | – | – | -.29 | 3 | 6642 |
| Conscientiousness and substance use | Hakulinen, et al. (2015) | – | 0.99 | .00[a] | 8 | 24454 |
| | Kotov et al. (2010) [38] | -1.1 | – | -.44[a] | 25 | 23811 |
| Extroversion and neuroticism | Lipnevich et al. (2017) [103] | – | – | -.16 | 27 | 9819 |
| | Steel et al. (2008) [104] | – | – | -.33 | 57 | 20178 |
| | Ones (1993) [105] | – | – | -.19 | 710 | 440440 |
| Extroversion and openness | Lipnevich et al. (2017) [103] | – | – | .14 | 27 | 9819 |
| | Steel et al. (2008) [104] | – | – | .24 | 28 | 9912 |
| | Ones (1993) [105] | – | – | .17 | 418 | 252004 |
| Extroversion and depression | Kotov et al. (2010) [38] | -.62 | – | -.25[a] | 55 | 56823 |
| Extroversion and anxiety | Kotov et al. (2010) [38] | -1.02 | – | -.18[a] | 10 | 29065 |
| Extroversion and substance use | Hakulinen, et al. (2015) [41] | – | .91 | -.04[a] | 8 | 24454 |
| | Kotov et al. (2010) [38] | -.36 | – | -.16 | 49 | 12290 |
| Neuroticism and openness-to-experience | Lipnevich et al. (2017) [103] | – | – | -.07 | 27 | 9818 |
| | Steel et al. (2008) [104] | – | – | -.09 | 33 | 11682 |
| | Ones (1993) [105] | – | – | -.16 | 423 | 254937 |
| Neuroticism and depression | Kotov et al. (2010) [38] | 1.33 | – | .47[a] | 63 | 75229 |
| Neuroticism and anxiety | Kotov et al. (2010) [38] | 1.96 | – | .34[a] | 14 | 46244 |
| Neuroticism and substance use | Kotov et al. (2010) [38] | .97 | – | .36[a] | 58 | 68075 |
| Openness-to-experience and depression | Kotov et al. (2010) [38] | -.21 | – | -.08[a] | 26 | 24886 |
| Openness-to-experience and anxiety | Kotov et al. (2010) [38] | -.40 | – | -.09[a] | 4 | 10609 |
| Openness-to-experience and substance use | Hakulinen, et al. (2015) | – | .90 | -.04[a] | 8 | 24454 |
| | Kotov et al. (2010) [38] | -.16 | – | -.07[a] | 26 | 7709 |
| Depression and anxiety | Jacobson & Newman (2017) [106] | – | 2.55 | .34[a] | 38 | 38902 |
| Depression and substance use | Lai, Cleary, Sitharthan, & Hunt (2015) [107] | – | 2.42 | .32[a] | 34 | 504319 |
| | Conner, Pinquart, & Duberstein (2008) [108] | – | – | .08 | 7 | – |
| Anxiety and substance use | Lai et al. (2015) [107] | – | 2.11 | .27 | 31 | 504319 |

[a] Either converted from odds ratios or *d*-statistics.

Effect sizes reported in meta-analyses as Cohen's *d*-statistic or as OR required transformation to correlations. Transformation from *d* to *r* is commonplace but from OR to *r* is not as straightforward. When the case/treatment and control groups are dissimilar in size, as they were in the meta-analyses coded here, it is recommended [113] that the values be transformed as an approximation of the tetrachoric correlation [114] which is preferred over a Pearson transformation [115] which requires equal marginal probabilities. Alas, none of the studies that computed meta-analytic effect sizes as odds ratios reported their outcomes as continuous variables, or had similar size samples in the cells of the contingency table, or had equal marginal probabilities so the OR values were transformed into an approximation of the tetrachoric correlation using the formula [114] below:

$$r = [OR^{3/4} - 1]/[OR^{3/4} + 1] \qquad [1]$$

### Creation of correlation matrices for input

As can be seen in the online supplementary materials, the 45 unique off-diagonal cells of the matrix had between 1 and 3 meta-analyses each that were coded. Thus, we had an impossibly

large 120,932,352 different possible permutations of the input matrix. We ran three different matrices comprised of meta-analytic correlations that were not just a randomly assembled set of correlations serving as one of the over 120 million possible combinations. Our matrices were purposeful and were (a) the most recent meta-analyses in the cells under the assumption that these would be the most comprehensive, (b) the meta-analyses that had the largest individual number of subjects or largest $n$ in each cell, and (c) the meta-analyses that had the largest number of samples used or largest $k$ per cell. The other potentially viable but randomly assembled candidates of possible input matrices were not examined here. Because the model input effect sizes for the relationship between anxiety and substance use and between depression and substance use were meta-analyzed on longitudinal data some inference of causality can be inferred for these two pairs of variables. Personality is omnipresent, conflict is not usually episodic, and mental disorders usually take time to develop so they all tend to co-occur and temporal precedence is difficult to establish between them. However, our model and data imply that anxiety and depression temporally precede substance use thus lending some causality to an otherwise correlational model.

## Data analysis

The hypothesized model in Fig 2 was tested using each of the three aforementioned correlation matrices as input to Mplus 8.2 SEM software [116]. The maximum likelihood estimator function was used on the summary data. Unstandardized paths were not computed because the use of correlations without standard deviations or means for model variables served as the data input. Following well-known recommendations [93], the harmonic mean was used to determine the sample size of each of the three input matrices. Bootstrapping was used to estimate confidence intervals around path coefficients. The disturbance terms for work-to-family conflict and family-to-work conflict were allowed to correlate. Separately, the disturbance terms for anxiety and depression were also allowed to correlate. We then evaluated the indirect effects by constructing bias-corrected 95% confidence intervals [117, 118].

## Results

The input matrix comprised of the most recently published meta-analytic correlations using the harmonic mean of 8942, shown in Table 2, yielded better fit [$\chi^2$ = 744.69***, $df$ = 7, SRMR = .04, CFI = .93, RMSEA = .109 [90% C.I. = .102; .115] than the models run on the two other input matrices using different respective harmonic means [model with meta-analytic largest $n$: $\chi^2$ = 2769.49***, $df$ = 7, SRMR = .06, CFI = .84, RMSEA = .183 [90% C.I. = .177; .189] and the model with meta-analytic largest $k$: $\chi^2$ = 2506.86***, $df$ = 7, SRMR = .07, CFI = .85, RMSEA =

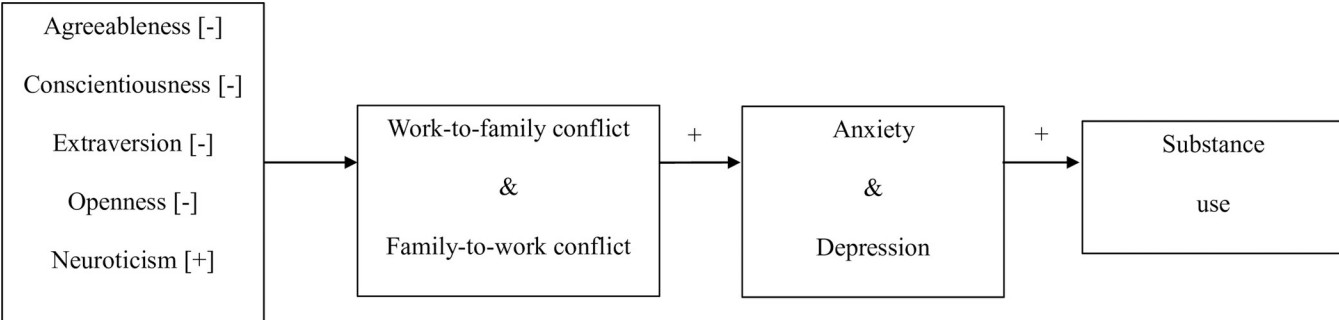

**Fig 2. Hypothesized model.** The model of some antecedents and outcomes of work-family and family-work conflict.

**Table 2. Input matrix comprised of correlations[a] and standard deviations from most recent meta-analyses.**

| Variables | 1. | 2 | 3. | 4. | 5. | 6. | 7. | 8. | 9. | 10. |
|---|---|---|---|---|---|---|---|---|---|---|
| 1. Work-to-family conflict | – | .1732 | .0548 | .1049 | .1732 | .1049 | .0837 | .0288 | .0288 | .0460 |
| 2. Family-to-work conflict | 0.41 | – | .0707 | .0894 | .0316 | .1049 | .0707 | .0371 | .1436 | .0516 |
| 3. Agreeableness | -0.17 | -0.19 | – | .1900 | .2000 | .2700 | .1900 | .2448 | .3347 | .0153 |
| 4. Conscientiousness | -0.16 | -0.20 | 0.32 | – | .1200 | .2500 | .2300 | .1618 | .1267 | .0128 |
| 5. Extroversion | -0.09 | -0.07 | 0.20 | 0.21 | - | .2500 | .1400 | .2659 | .3123 | .0178 |
| 6. Neuroticism | 0.31 | 0.27 | -0.11 | -0.17 | -0.16 | – | .1700 | .2418 | .2079 | .0153 |
| 7. Openness | -0.02 | -0.05 | 0.21 | 0.16 | 0.22 | -0.07 | – | .2631 | .2508 | .0127 |
| 8. Depression | 0.23 | 0.22 | -0.06 | -0.36 | -0.25 | 0.47 | -0.08 | – | .1453 | .0479 |
| 9. Anxiety | 0.14 | 0.19 | 0.05 | -0.29 | -0.18 | 0.34 | -0.09 | 0.34 | – | .0180 |
| 10. Substance use | 0.08 | 0.10 | 0.03 | 0.00 | -0.04 | 0.36 | -0.04 | 0.32 | 0.27 | – |

[a] Correlations are on bottom left side of matrix below the diagonal and standard deviations are on top right side above the diagonal. Harmonic mean = 8942. All non-zero correlations are significant at $p < .001$.

.177 [90% C.I. = .171; .183] and was the focus of further analysis. The SRMR indicates excellent fit [119], the CFI indicates good fit [120], and the RMSEA shows poor fit [121] for the model using the most recent meta-analytic correlations as input. Overall, that model ran can be said to show adequate fit to the data and better fit than the models run on the other two input matrices. The fit indices and path coefficients detailed below are likely to be quite robust in that the average number of studies used in the meta-analyses in each cell was 20.56 thus exceeding the minimum recommendation of 10 per cell [91].

In post hoc tests using the FIMSEM technique [95] on our best fitting input matrix, which required the correlation matrix as well as the standard deviation matrix, we ran 500 bootstrapped samples. We found that the bootstrapped path coefficients were close approximations of our model test but that the fit indices were not reproduced well [102] nor were they meaningful. The interested reader can use the correlations and standard deviations in Table 2 to reproduce these FIMASEM results.

## Simple mediation results

Hypotheses 1 through 6 predicted simple mediation. Results for these tests can be found in Table 3. Hypotheses 1a through 1e all involved WFC as a mediator between each of the Big Five traits and anxiety. Because the path between WFC and anxiety was .00, the indirect paths from these traits through WFC to anxiety were also .00. Therefore, none of these five hypotheses were supported.

Hypotheses 2a through 2e all involved FWC as a mediator between the Big Five traits and anxiety. Hypothesis 2a, that FWC mediates the linkage from agreeableness to anxiety and was supported. Hypothesis 2b, that FWC mediates the linkage from conscientiousness to anxiety also was supported. Hypotheses 2c and 2d, about extroversion and openness respectively, were not supported. Finally, Hypothesis 2e, that FWC mediates the linkage from neuroticism to anxiety was supported.

Hypotheses 3a through 3e all involved WFC as a mediator between each of the Big Five traits and depression. Hypothesis 3a, that WFC mediates the linkage from agreeableness to depression was supported as was Hypothesis 3b about the linkage from conscientiousness to WFC to depression. and. Hypotheses 3c about extroversion was not supported nor was Hypothesis 3d about the linkage from openness to WFC to depression because the direct of the effect was opposite that predicted. Finally, Hypothesis 3e about the linkage from neuroticism to WFC to depression was supported.

**Table 3. Results for simple mediation.**

| Hypothesis | Path | Effect | 95% CI | |
|---|---|---|---|---|
| | | | Lower | Upper |
| 1a | Agreeableness->WFC->Anxiety | .00 | -.003 | .002 |
| 1b | Conscientiousness->WFC->Anxiety | .00 | -.002 | .002 |
| 1c | Extroversion->WFC->Anxiety | .00 | .000 | .000 |
| 1d | Openness->WFC->Anxiety | .00 | -.001 | .001 |
| 1e | Neuroticism->WFC->Anxiety | .00 | -.006 | .006 |
| 2a | Agreeableness->FWC->Anxiety | -.01*** | -.016 | -.009 |
| 2b | Conscientiousness-> FWC ->Anxiety | -.12*** | -.015 | -.008 |
| 2c | Extroversion-> FWC ->Anxiety | .00 | .000 | .004 |
| 2d | Openness-> FWC ->Anxiety | .00 | -.001 | .003 |
| 2e | Neuroticism-> FWC ->Anxiety | .02*** | .017 | .028 |
| 3a | Agreeableness->WFC->Depression | -.01*** | -.009 | -.004 |
| 3b | Conscientiousness->WFC-> Depression | -.004*** | -.006 | -.002 |
| 3c | Extroversion->WFC-> Depression | .00 | -.002 | .000 |
| 3d | Openness->WFC-> Depression | .002** | .001 | .003 |
| 3e | Neuroticism->WFC-> Depression | .02*** | .009 | .020 |
| 4a | Agreeableness->FWC-> Depression | -.01*** | -.009 | -.004 |
| 4b | Conscientiousness-> FWC -> Depression | -.01*** | -.009 | -.004 |
| 4c | Extroversion-> FWC -> Depression | .00 | .000 | .002 |
| 4d | Openness-> FWC -> Depression | .00 | -.001 | .002 |
| 4e | Neuroticism-> FWC -> Depression | .01*** | .008 | .017 |
| 5a | WFC->Anxiety->Substance use | .00 | -.004 | .004 |
| 5b | WFC->Depression->Substance use | .04*** | .008 | .019 |
| 6a | FWC->Anxiety->Substance use | .02*** | .013 | .021 |
| 6b | FWC->Depression->Substance use | .01*** | .008 | .018 |

*Note*: WFC = work-to-family conflict; FWC = family-to-work conflict; CI = confidence interval.

***$p < .001$.

Hypotheses 4a through 4e all involved FWC as a mediator between the Big Five traits and depression. Hypothesis 4a, which predicted a mediating effect of FWC for the agreeableness to depression relationship, was supported as was Hypothesis 4b, that predicted a mediating effect of FWC for the conscientiousness to depression relationship. Hypotheses 4c [extroversion] and Hypothesis 4d [openness] were not supported. Finally, Hypothesis 4e, which predicted a mediating effect for FWC for the neuroticism to depression relationship, was supported.

Hypotheses 5a and 5b were about the linkage from WFC through anxiety and depression, respectively, to substance use. Because the path between WFC and anxiety was .00, Hypothesis 5a was not supported. However, the indirect path through depression was significant offering support for Hypothesis 5b. Hypotheses 6a and 6b were about the linkages from FWC through both anxiety and depression to substance use. The indirect effects through anxiety and depression on the relationship between FWC and substance use were both significant providing support for Hypotheses 6a and 6b.

## Sequential mediation results

The results for our tests of sequential mediation for Hypotheses 7a-10e can be found in Table 4. Hypotheses 7a-e predicted sequential mediation between each of the Big Five personality traits one at a time and substance use through WFC and anxiety. None of these indirect

**Table 4. Results for sequential mediation.**

| Hypothesis | Path | Effect | 95% CI[a] | |
|---|---|---|---|---|
| | | | Lower | Upper |
| 7a | Agreeableness->WFC[b]->Anxiety->Substance Use | .000 | .000 | .000 |
| 7b | Conscientiousness->WFC->Anxiety->Substance Use | .000 | .000 | .000 |
| 7c | Extroversion->WFC->Anxiety->Substance Use | .000 | .000 | .000 |
| 7d | Openness->WFC->Anxiety->Substance Use | .000 | .000 | .000 |
| 7e | Neuroticism->WFC->Anxiety->Substance Use | .000 | -.001 | .001 |
| 8a | Agreeableness->FWC[c]->Anxiety->Substance Use | -.002*** | -.003 | -.002 |
| 8b | Conscientiousness-> FWC ->Anxiety->Substance Use | -.002*** | -.003 | -.001 |
| 8c | Extroversion-> FWC ->Anxiety->Substance Use | .000 | .000 | .001 |
| 8d | Openness-> FWC ->Anxiety->Substance Use | .000 | .000 | .001 |
| 8e | Neuroticism-> FWC ->Anxiety->Substance Use | .004*** | .003 | .005 |
| 9a | Agreeableness->WFC->Depression->Substance Use | -.002*** | -.002 | -.001 |
| 9b | Conscientiousness->WFC-> Depression->Substance Use | -.001*** | -.002 | -.001 |
| 9c | Extroversion->WFC-> Depression->Substance Use | .000 | .000 | .000 |
| 9d | Openness->WFC-> Depression->Substance Use | .001** | .000 | .001 |
| 9e | Neuroticism->WFC-> Depression->Substance Use | .004*** | .002 | .005 |
| 10a | Agreeableness->FWC-> Depression->Substance Use | -.002*** | -.002 | -.001 |
| 10b | Conscientiousness-> FWC -> Depression->Substance Use | -.002*** | -.002 | -.001 |
| 10c | Extroversion-> FWC -> Depression->Substance Use | .000 | .000 | .001 |
| 10d | Openness-> FWC -> Depression->Substance Use | .000 | .000 | .000 |
| 10e | Neuroticism-> FWC -> Depression->Substance Use | .003*** | .002 | .004 |

[a] CI = confidence interval.

[b] WFC = work-to-family conflict.

[c] FWC = family-to-work conflict.

*** $p < .001$.

effects were significant as the path between WFC and anxiety was .00, so Hypotheses 7a-e were not supported. Hypotheses 8a-e predicted sequential mediation between each of the Big Five personality traits one at a time and substance use through FWC and anxiety. Hypothesis 8a (agreeableness), 8b (conscientiousness), and 8e (neuroticism) were all supported. However, Hypotheses 8c (extroversion) and 8d (openness) were not supported. Hypotheses 9a-e predicted sequential mediation between each of the Big Five personality traits one at a time and substance use through WFC and depression. The hypotheses for agreeableness (9a), conscientiousness (9b), and neuroticism (9e) were all supported while Hypotheses 9c (extroversion) was not. Although significant results were found for Hypothesis 9d (openness), the indirect effect was not in the predicted direction and the confidence interval contained zero, therefore Hypothesis 9e also was not supported. Finally, Hypotheses 10a-e predicted sequential mediation between each of the Big Five personality traits one at a time and substance use through FWC and depression. These results followed the same pattern as the sequential mediation for anxiety and FWC in that Hypothesis 10a (agreeableness), 10b (conscientiousness), and 10e (neuroticism) were all supported while Hypotheses 10c (extroversion) and 10d (openness) were not. See Fig 3 for the statistical results of the mega-meta path analysis model test.

## Discussion

The goal of our study was to test part of the nomological network of WFC and FWC via a simultaneous investigation of some of their antecedents and outcomes that integrated the

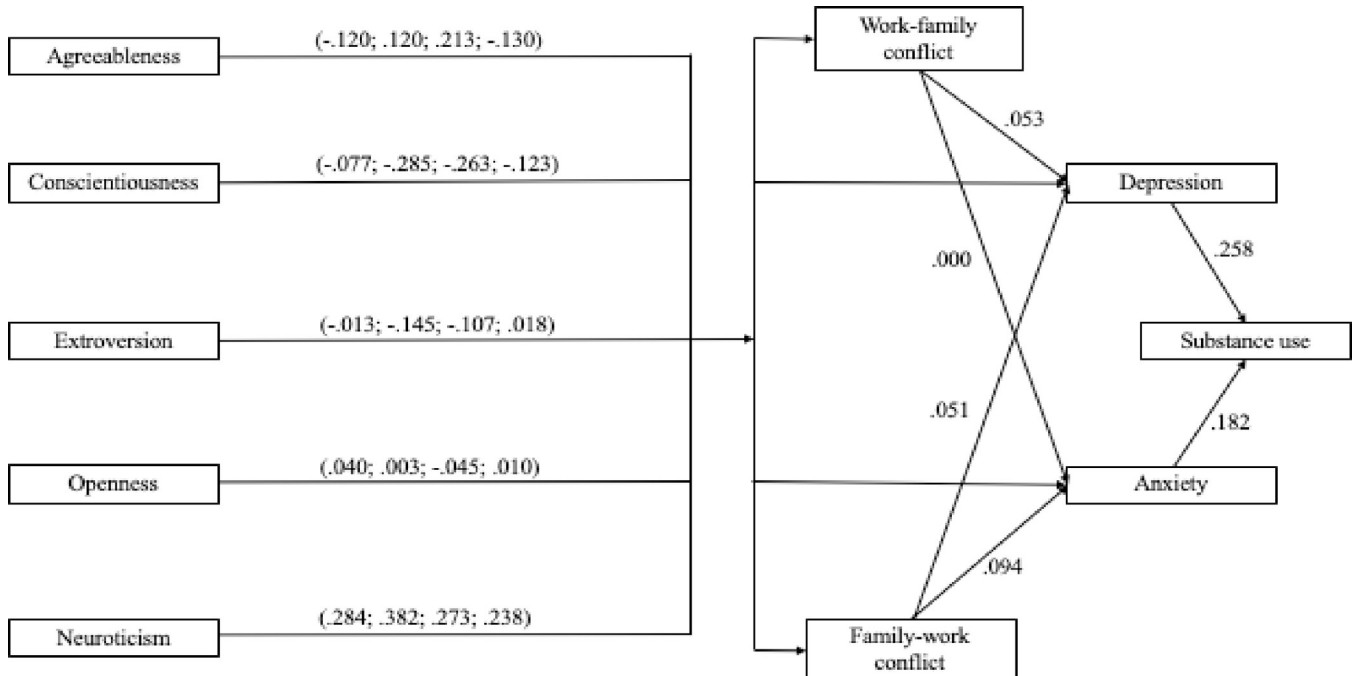

**Fig 3. Model of statistical results.** Path coefficients resulting from mega-meta path analysis. *Note.* Values in parentheses indicate direct paths from each trait to work-to-family conflict, depression, anxiety, and family-to-work conflict in that order. Paths > .020 are statistically significant. For clarity, the intercorrelations between the Big Five traits and the correlations between the disturbance terms for the endogenous variables are omitted.

loosely connected fields of management, personality, clinical psychology, and public health research. Specifically, we examined the antecedent role of the Big Five personality traits in the prediction of the maladaptive behavior of substance use through the sequential mediation of WFC, FWC, clinical anxiety, and clinical depression. Consistent with COR theory [24], we hypothesized and found meaningful sequential mediation effects among the personality traits of agreeableness, conscientiousness, and neuroticism, with both WFC and FWC, and depression in the prediction of substance use. As expected, the results suggest that agreeableness and conscientiousness are key resources that one can utilize to reduce the incompatible role conflict between work and family domains [49, 51], resulting in lower depression and further mitigating substance use. On the other hand, neuroticism greatly hinders one's use of resources that then increase the propensity for WFC/FWC as well as contributes to depression and substance use. Due to dysfunctional emotional adjustments, sensitivity, and vulnerability to negative experiences [54, 59, 89], neurotic employees have fewer resources with which to manage incompatible work and family demands [46], leading to higher WFC, which in turn increases depression and substance use.

Regarding the mediating role of anxiety, family-to-work conflict but not work-to-family conflict, was a significant mediator linking agreeableness, conscientiousness, and neuroticism with anxiety and substance use. This is probably because of the different contexts of work-to-family conflict and family-to-work conflict. As work-to-family conflict is the interference from work into the family domain (e.g., having to reply to emails or work on projects during family time) that may require additional contextual resources, such as family member support [122] and supervisor support [16], to lower the levels of anxiety and substance use. As for neuroticism, the findings demonstrate that its indirect effect on substance use is channeled through work-to-family conflict, family-to-work conflict, anxiety, or depression to impact substance

use. Moreover, our findings indicate that extroversion and openness-to-experience have weak or nonsignificant effects on substance use through WFC and anxiety or through depression. Taken together, this study suggests that personality, as a stable key resource [29], can facilitate or hinder WFC/FWC and mental health. Positive, or adaptive, traits like agreeableness and conscientiousness provide strength to overcome conflict in life and work. Negative, or mal-adaptive, traits like neuroticism detract from one's ability to weather the storm of conflict and play a direct role in increasing conflict and mental health disorders. For some employees, adaptive traits can serve as a reliable buttress to the struggles of life at work and at home and the sometimes-inevitable blurring of the boundaries between the two.

## Study strengths, limitations and future directions

This study has several strengths. First, we used large sample sizes based upon meta-analyses of hundreds of different primary published studies thus helping ameliorate second order sampling error. The overall number of participants in the meta-analyses of the relationships used to create our best fitting input correlation matrix was 1,213,933. However, it is likely that some participants were counted more than once as some correlations in each input matrix came from the same study and presumably were calculated from the same primary study participants. Second, most of the meta-analytically derived correlations were corrected for attenuation by the original meta-analysts thus more closely approximating population parameters. Third, the meta-analyses were collected from many top journals covering general psychology, general management, organizational behavior, human resource management, clinical psychology, and epidemiology. Finally, using COR theory [24], in conjunction with the self-medication hypothesis, we were able to theoretically integrate the left-hand side (i.e. antecedents) with the right-hand side (i.e. outcomes) of the WFC/FWC network that are usually explored separately. Integrating these seemingly disparate meta-analytic correlations from different fields of study serves as an effective comprehensive explanation for the modeled relationships. These strengths lend credibility to the findings reported here.

Our study is not without limitations. The socio-demographic variables such as type of work and organizational tenure related to work were not, and largely could not be, included in the study. The body of literature around the relationship between them and WFC/FWC is growing but meta-analyses do not exist for most work-related socio-demographic variables with WFC/FWC and definitely do not exist for the relationship between them and the two disorders and substance use. Like with meta-analyses in general there are many judgment calls [123] with which to contend when using a mega-meta path analytic framework. For example, we are limited by the way other scholars coded their variables in their meta-analyses. Given that some variables may have been dichotomized or truncated in other ways, future primary study researchers could include more nuanced measures of anxiety and depression, thus providing more accurate meta-analytic insights. Additionally, the use of discrete categories to describe mental disorders likely attenuated some model relationships. If only the severest of cases can be categorized as clinically anxious or depressed, then the relationship between these disorders and the other variables in the model were likely weaker here than they could be if anxiety and depression were measured on continuous scales. Thus, our model may actually underreport the strength of these relationships. We were also not able to test some seemingly logical model relationships because published meta-analyses could not be found between all such variables. For example, although we focused on the clinical forms of depression and anxiety we could not include substance use disorder in our model because meta-analyses do not exist between that construct and all nine other variables in our model. Instead we focused on sub-clinical substance use because that is what was available.

The mega-meta path analytic technique can be limited by the availability of existing meta-analyses. Another potential weakness is that of effect size heterogeneity [101] in that there were quite a range of meta-analytic values reported for the same relationship in different meta-analyses. For example, one meta-analysis [41] reported an odds ratio between agreeableness and substance use that was transformed into a correlation of .03. For that same relationship, another meta-analysis [38] reported a *d*-statistic that was transformed into a correlation of -.27. The heterogeneity of these effect sizes suggests one other related limitation: the transformation of odds ratios is only an approximation of a tetrachoric correlation and not a direct Pearson correlation of continuous measures. Additionally, we only tested three of the over 120 million possible permutations of the input correlation matrix that were possible. It is highly likely that more than one of the other possible matrices would fit our model even better than those which we tested.

## Conclusion and practical implications

In conclusion, we designed this research to connect some of the antecedents and outcomes of the nomological network surrounding work-to-family and family-to-work conflict. These two forms of conflict are mediators of the relationship between personality and mental health. We found that the three personality traits of agreeableness, conscientiousness, and neuroticism worked through the mediating mechanisms of work-to-family conflict and family-to-work conflict with depression as well as through family-to-work conflict and anxiety to play mediating roles in their relationship with substance use. Thus, this research supports the notion that the work-family interface may play a critical role in understanding why employees experience adverse mental health outcomes and engage in substance use.

We suggest that organizations take kinder note of employees facing difficulties at work or at home and that they appropriate funding for employee assistance programs to address mental health and substance use problems. Many of the types of issues giving rise to $400 billion in annual losses to businesses in the U.S. [77] are the result of mental health issues and/or substance use and abuse. The loss to businesses is not only financial but interpersonal as the spill-over effect of idiosyncratic problems of one worker may unintentionally affect another worker as the cascade of despair infects the organization. Given the rise in interest in personality testing by managers in the workplace we recommend that those employees with an identifiably aberrant combination of traits (e.g. extremely low agreeableness and extremely high neuroticism) be given an opportunity to receive wise counsel from supervisors regarding interpersonal interactions with coworkers and customers. However, the relative unmalleability of personality traits makes such trait levels particularly difficult to ameliorate but the encouragement of self-awareness can likely be beneficial. To facilitate such self-awareness, we recommend training in emotional intelligence which helps employees build skills devoted to helping them be aware of and manage the emotions of themselves and others. The goal of a business organization should be not only to earn a profit but to develop its people to their fullest human capacity.

## Supporting information

**S1 Checklist. PRISMA checklist.**
(DOCX)

## Acknowledgments

A previous version of this manuscript was accepted in reduced form and with a different title to be presented at the annual conference of the Academy of Management in Vancouver, British Columbia, Canada in 2020. However, the conference was cancelled due to COVID-19.

## Author Contributions

**Conceptualization:** Brian K. Miller.

**Data curation:** Maggie Wan, Dawn Carlson, K. Michele Kacmar, Merideth Thompson.

**Formal analysis:** Brian K. Miller, K. Michele Kacmar.

**Funding acquisition:** Brian K. Miller.

**Methodology:** Brian K. Miller.

**Project administration:** Brian K. Miller.

**Writing – original draft:** Brian K. Miller, Maggie Wan.

**Writing – review & editing:** Dawn Carlson, K. Michele Kacmar, Merideth Thompson.

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
