## [Decision Letter · Decision Letter 0]

12 Jan 2022

PONE-D-21-39844Antecedents and Outcomes of Work-Family Conflict:  A Mega-Meta Path AnalysisPLOS ONE

Dear Dr. Brian K. Miller,

Thank you for submitting your manuscript to PLOS ONE. After careful consideration, we feel that it has merit but does not fully meet PLOS ONE’s publication criteria as it currently stands. Therefore, we invite you to submit a revised version of the manuscript that addresses the points raised during the review process.

We look forward to receiving your revised manuscript.

Kind regards,

Rogis Baker, Ph.D

Academic Editor

PLOS ONE

Journal Requirements:

Reviewers' comments:

Reviewer's Responses to Questions

**Comments to the Author**

1. Is the manuscript technically sound, and do the data support the conclusions?

Reviewer #1: Yes

Reviewer #2: Partly

2. Has the statistical analysis been performed appropriately and rigorously? 

Reviewer #1: Yes

Reviewer #2: I Don't Know

3. Have the authors made all data underlying the findings in their manuscript fully available?

Reviewer #1: Yes

Reviewer #2: Yes

4. Is the manuscript presented in an intelligible fashion and written in standard English?

Reviewer #1: Yes

Reviewer #2: Yes

5. Review Comments to the Author

Reviewer #1: I appreciate the opportunity to review this article. The authors conducted a mega-meta analytic path analysis examining the relationships among Big Five traits, work-to-family conflict and family-to-work conflict, anxiety and depression, and substance use.

In my opinion the objective is relevant and the selection of variables and the theoretical models are also well founded.

The methodology is complex and allows responding to the objectives set.

However, there are some doubts related to the definition of some variables and the methodology used.

- I think that the constructs “work-to-family conflict and family-to work conflict” should be defined more extensively in the introduction.

- In the discussion some traits are considered as adaptive and one trait as maladaptive. However, the Big Five Traits Model is a normal personality model. According to the model, no trait is maladaptive in itself, but it will depend on the context or environment in which the person is and the influence of other variables. Is neuroticism considered a maladaptive trait in the context of the interpretation of the results?

- I understand that only clinical cases of anxiety and depression with a clinical diagnosis have been considered. Is that correct? Could any diagnostic category be considered of both types of disorders?

- When describing the Article Inclusion Criteria, it is not clear what the reasons were for excluding the 115 articles from the last step of Figure 1.

- Although I am not an expert in mega-meta

analytic path analysis, I think that as described by the authors, the working method can be followed without difficulty.

- I think that within the limitations it should be included that the influence of sociodemographic variables and variables related to work (type of work, years of dedication, occupational risks, temporality) have not been considered.

- Practical implications should be included. According to the results, what strategies could be implemented to reduce work-to-family conflict and family-to-work conflict? How to prevent them? Could it be predicted through the personality profile?

In summary, I think it is an interesting study that uses a complex methodology; however, there are some aspects that should be clarified.

Reviewer #2: Thank you very much for this opportunity to read this interesting study. Although, the paper is highly interesting and worth publishing, I have some remarks and questions.

Introduction

I would have wanted stronger theoretical and conceptual background in the introduction.

For example, to clarify what is the difference of WFC and FWC? And why they need to be considered separately?

Why anxiety, depression and substance use were selected as outcomes?

Could the relationship go other way around? I mean could it be that anxiety, depression and/or substance use affect WFC and/or FWC?

Results

In table 3, 2e Neuroticism-> FWC-> Anxiety, should the effect be positive, instead of -.02***? The 95% confidence intervals are positive

Is 3a in the same table correct effect?

Now there are quite many analyses. Did you use e.g. Bonferroni correction for the p-values?

I was wondering, have you tested e.g. to combine WFC and FWC to a common conflict measure and run the analyses with it? As e.g. Byron 2005 (your reference number 10) shows that coping styles and skills have quite similar relations with both WFC and FWC, and I think personality traits might behave similarly. What comes to your reference number 11, it considers positive side of work-family interface, not conflict, what you concentrate on, so I don’t think it is the best one for your purposes.

I would ask the same with anxiety and depression, such as have you tried to combine these to general mood or mental health measure and run the analyses with it?

Do these combinations affect the effect sizes? Or to the model fit indexes?

RMSEA shows poor fit, is that a reason to be concerned?

6. PLOS authors have the option to publish the peer review history of their article (what does this mean?). If published, this will include your full peer review and any attached files.

Reviewer #1: No

Reviewer #2: No

---

## [Author Response · Author response to Decision Letter 0]

18 Jan 2022

Response to Reviewers of PLOS-D-21-39844

Reviewer #1 Comments: 

1. I appreciate the opportunity to review this article. The authors conducted a mega-meta analytic path analysis examining the relationships among Big Five traits, work-to-family conflict and family-to-work conflict, anxiety and depression, and substance use. In my opinion the objective is relevant and the selection of variables and the theoretical models are also well founded. The methodology is complex and allows responding to the objectives set. However, there are some doubts related to the definition of some variables and the methodology used.

Authors' Response: We thank the reviewer for these comments and address them one-by-one below in a numbered fashion. 

2. I think that the constructs “work-to-family conflict and family-to work conflict” should be defined more extensively in the introduction.

Authors' Response: Thank you for your suggestion. We elaborate as suggested on the definitions of work-to-family conflict and family-to-work conflict on newly inserted lines 46-60 into the second paragraph of our paper: 

"With the development of valid instruments [2, 5, 6, 7] research on the antecedents, correlates, and outcomes of work-to-family conflict (WFC) and family-to-work conflict (FWC) has grown in number and type. The important directionally different flow of conflict between these two major domains of life [8] results in WFC and FWC being only moderately positively correlated at .41 [9]. It is notable that work-family conflict takes unique bi-directional forms of WFC and FWC. The former is defined as when the work demands interfere with performing family responsibilities, while the latter is defined as when family demands interfere with the performance of work duties [4]. Examples of the former include work demands like forced overtime and dealing with rude customers. Examples of the latter include family demands like caring for sick children and marital strife. These two directions of conflict have enough unique variance to act as separate and stand-alone constructs because they only overlap by about 16% [9]. A large body of literature [2, 5, 9] has documented that WFC and FWC are theoretically distinct, are predicted by different factors, and result in different consequences [2, 5, 9]. Because these two forms of conflict often relate differently to other variables [10, 11] they are not interchangeable."

3. In the discussion some traits are considered as adaptive and one trait as maladaptive. However, the Big Five Traits Model is a normal personality model. According to the model, no trait is maladaptive in itself, but it will depend on the context or environment in which the person is and the influence of other variables. Is neuroticism considered a maladaptive trait in the context of the interpretation of the results?

Authors' Response: We appreciate this request for clarification and hope that our response is not pedantic in any way. Our understanding of the Big Five follows. The traits in the model are construed as a spectrum which ranges from very low to very high levels of each trait. Some of the traits are known by their polar opposite endpoint name. For example, agreeableness could be referred to (but rarely is) as "disagreeableness" if the need arose. Such a need might be in the explanation of its positive relationship with Anti-Social Personality Disorder (ASPD). Alternatively, ASPD is negatively correlated with trait agreeableness. 

The negative relationship between agreeableness scores and ASPD would likely make agreeableness adaptive in some cases. Low scores on agreeableness, or alternatively construed as high scores on disagreeableness, would make it somewhat maladaptive. It is the same trait, it just matters from which direction it is measured. We are quite certain that the reviewer already knows this but wanted to make sure that our understanding of the Big Five traits matches that of the reviewer. 

Now we directly address the reviewer's issue. For many years, the trait of Emotional Stability was known as Neuroticism which is likely a holdover of Eysenck's two factor model of personality. When the trait is measured such that high scores on it indicate high levels of neuroticism and therefore low levels of emotional stability then in some contexts it can be maladaptive in its behavioral, cognitive, and affective outcomes. In the studies coded in the meta-analyses gathered for use in this paper the meta-analysts tended to measure that Big Five trait as neuroticism which is negatively related to most favorable outcomes and positively related to most maladaptive outcomes like WFC, FWC, depression, anxiety, and substance use. We found that in the context of the work-family interface, neuroticism is strongly related to all five. We consider this to be evidence of its maladaptive nature. 

Methodologically, we would only need to reverse the sign of the model relationships and we would have results of an opposite but equal magnitude. In that case all Big Five Traits in our model would be negatively related (adaptive) to the various hypothesized outcomes in our model. We are happy to go back and reverse the signs to accomplish this but theoretically it weakens our argument about Conservation of Resources Theory which helps the reader understand the importance of neuroticism in the prediction of maladaptive outcomes like depression, anxiety, and substance use. We hope this helps explain our thinking and we thank the reviewer for this comment. 

4. I understand that only clinical cases of anxiety and depression with a clinical diagnosis have been considered. Is that correct? Could any diagnostic category be considered of both types of disorders?

Authors' Response: Yes, only clinically diagnosed cases were considered because that was all that was available in the published meta-analyses used as part of the synthetic input correlation matrix for this study. In those meta-analyses, different meta-analytically derived values for different directional influences of WFC and FWC on depression and anxiety were available. Meta-analytic effect sizes do exist for depression and anxiety in the antecedent prediction of WFC and FWC. To our knowledge none considered sub-clinical levels of depression and anxiety. Our study can only be thought of as truly cross-sectional we opted for the meta-analytic values of WFC and FWC as truly and timely precursors of depression and anxiety to lend a modicum of causality to the model. However, these are not consistent with the field of research surrounding WFC/FWC which developed along the lines of clear and distinct antecedents of conflict and clearly theoretical outcomes of conflict. We noted our lack of the ability to infer causality as a limitation in the original submitted version of our study which we still include and can be found on the newly numbered lines 391-396 and make sure to limit any notion of causality. 

5. When describing the Article Inclusion Criteria, it is not clear what the reasons were for excluding the 115 articles from the last step of Figure 1.

Authors' Response: Thanks for the opportunity to explain this. We excluded those studies because they were meta-analyses on relationships other than the 45 necessary for input into our synthetic correlation matrix. We have changed the verbiage in two of the boxes in the flow chart to be more transparent. Instead of one box stating "Records excluded (n = 4087)" we rewrote it as "Non-meta-analyses excluded (n = 4087)". These records were excluded because they merely referenced a meta-analysis and were not meta-analyses in and of themselves. To directly address the reviewer's point, we also rewrote "Full text articles excluded (n = 115)" as "Irrelevant meta-analyses excluded (n = 115)". In our search we uncovered many meta-analyses that only referenced WFC and/or FWC and the Big Five or the psychological outcomes in our model. These 115 studies did not meta-analyze any of our focal relationships. We also found a typographical error in the text on line 345 where we deleted "606" and typed in "1039" to match the flow chart. 

6. Although I am not an expert in mega-meta analytic path analysis, I think that as described by the authors, the working method can be followed without difficulty.

Authors' Response: Thanks very much! It is indeed a complicated method so we tried very hard to make it understandable. 

7. I think that within the limitations it should be included that the influence of sociodemographic variables and variables related to work (type of work, years of dedication, occupational risks, temporality) have not been considered.

Authors' Response: We thank the reviewer for this suggestion. We added the following to lines 543-548: 

" The socio-demographic variables such as type of work and organizational tenure related to work were not, and largely could not be, included in the study. The body of literature around the relationship between them and WFC/FWC is growing but meta-analyses do not exist for most work-related socio-demographic variables with WFC/FWC and definitely do not exist for the relationship between them and the two disorders and substance use."

8. Practical implications should be included. According to the results, what strategies could be implemented to reduce work-to-family conflict and family-to-work conflict? How to prevent them? Could it be predicted through the personality profile?

Authors' Response: We thank the reviewer for this opportunity to add to our paper. On lines 586 -601 we have added the following: 

"We suggest that organizations take kinder note of employees facing difficulties at work or at home and that they appropriate funding for employee assistance programs to address mental health and substance use problems. Many of the types of issues giving rise to $400 billion in annual losses to businesses in the U.S. [77] are the result of mental health issues and/or substance use and abuse. The loss to businesses is not only financial but interpersonal as the spill-over effect of idiosyncratic problems of one worker may unintentionally affect another worker as the cascade of despair infects the organization. Given the rise in interest in personality testing by managers in the workplace we recommend that those employees with an identifiably aberrant combination of traits (e.g. extremely low agreeableness and extremely high neuroticism) be given an opportunity to receive wise counsel from supervisors regarding interpersonal interactions with coworkers and customers. However, the relative unmalleability of personality traits makes such trait levels particularly difficult to ameliorate but the encouragement of self-awareness can likely be beneficial. To facilitate such self-awareness, we recommend training in emotional intelligence which helps employees build skills devoted to helping them be aware of and manage the emotions of themselves and others. The goal of a business organization should be not only to earn a profit but to develop its people to their fullest human capacity."

We hope that we did not get too flowery in our language but we feel passionate about this set of implications and hope that they meet the reviewer's expectations. 

9. In summary, I think it is an interesting study that uses a complex methodology; however, there are some aspects that should be clarified.

Authors' Response: We thank the reviewer for the insightful comments and suggestions and hope that we have adequately addressed them. 

Reviewer #2 Comments: 

1. Thank you very much for this opportunity to read this interesting study. Although, the paper is highly interesting and worth publishing, I have some remarks and questions.

Authors' Response: We thank the reviewer for these suggestions, comments, and concerns and address them point-by-point below. 

2. Introduction. I would have wanted stronger theoretical and conceptual background in the introduction. For example, to clarify what is the difference of WFC and FWC? And why they need to be considered separately? 

Authors' Response: Thank you. Per your suggestion, we clarified the differences of WFC and FWC in the introduction and also explained why the two dimensions of work-family conflict need to be considered separately. The revised sentences can be found on the newly inserted lines 46-60 in the second paragraph of our paper: 

"With the development of valid instruments [2, 5, 6, 7] research on the antecedents, correlates, and outcomes of work-to-family conflict (WFC) and family-to-work conflict (FWC) has grown in number and type. The important directionally different flow of conflict between these two major domains of life [8] results in WFC and FWC being only moderately positively correlated at .41 [9]. It is notable that work-family conflict takes unique bi-directional forms of WFC and FWC. The former is defined as when the work demands interfere with performing family responsibilities, while the latter is defined as when family demands interfere with the performance of work duties [4]. Examples of the former include work demands like forced overtime and dealing with rude customers. Examples of the latter include family demands like caring for sick children and marital strife. These two directions of conflict have enough unique variance to act as separate and stand-alone constructs because they only overlap by about 16% [9]. A large body of literature [2, 5, 9] has documented that WFC and FWC are theoretically distinct, are predicted by different factors, and result in different consequences [2, 5, 9]. Because these two forms of conflict often relate differently to other variables [10, 11] they are not interchangeable."

Thus, following the existing literature, we consider the two variables separately in the study. 

3. Why anxiety, depression and substance use were selected as outcomes? Could the relationship go other way around? I mean could it be that anxiety, depression and/or substance use affect WFC and/or FWC?

Authors' response: We heavily considered running alternative models such as that. The fit of such a model would be exactly the same as our hypothesized model and the paths would likely only differ by the smallest of margins. It is true that mental disorders and substance use intuitively cause conflict at home and at work but the meta-analytic effect sizes gathered for use in this paper were computed on primary studies that largely had the intent of conflict being antecedent to the disorders. Thus, in the area of research around FWC and WFC depression, anxiety, and substance use have most often been studied as outcomes of FWC and WFC. Our paper is designed to strategically combine extant research from the growing field of WFC and FWC. We, in essence, combine the right half (i.e. antecedents ) and the left half (i.e. outcomes) of hundreds of studies of WFC and FWC based in our paper upon a unifying theory. Also, because our model can truly only be thought of as cross-sectional, because the nature of the meta-analytic results gathered for inclusion in our input matrix were calculated without any inference of directionality by the original meta-analysts. As we developed our model we scoured the databases for all primary studies of the antecedents and outcomes of WFC and FWC. We found a total of 56 different variables that neatly fell into either category. Alas, only the most oft-studied antecedent and outcome relationships had been meta-analyzed so we could not create a synthetic 56x56 matrix without relying on single primary studies. Additionally, some of the left-hand meta-analyses just did not combine well with some of the right-hand meta-analyses under any one or two particular theories. In essence, the field is a hodge-podge of loosely connected studies that share a central focus of WFC and FWC and largely based upon cross-sectional primary studies combined into non-directional effect sizes in meta-analyses. 

4. Results. In table 3, 2e Neuroticism-> FWC-> Anxiety, should the effect be positive, instead of -.02***? The 95% confidence intervals are positive. Is 3a in the same table correct effect?

Authors' Response: Oops. We apologize for this error in our results table. Thanks for the very keen eye for detail!

5. Now there are quite many analyses. Did you use e.g. Bonferroni correction for the p-values?

Authors' Response: We did not need to adjust for the many tests because we used structural equation modeling to test the model for fit and all model relationships for their relationships in one fell swoop. Consider, for example, a regression model with 2, or 5, or even 50 predictors. Each regression coefficient is already corrected for the propensity for Type I error in the model test. This might be a little off-note but there is some growing acknowledgement that when using meta-analytically derived effects that are corrected for measurement error as part of the meta-analysis one can refer to the path model test as a structural equation model test because such tests also correct for measurement error. Regardless of if we consider ours a structural equation model test or a path model test control of Type I error is an inherent part of the underlying algorithm. 

6. I was wondering, have you tested e.g. to combine WFC and FWC to a common conflict measure and run the analyses with it? As e.g. Byron 2005 (your reference number 10) shows that coping styles and skills have quite similar relations with both WFC and FWC, and I think personality traits might behave similarly. What comes to your reference number 11, it considers positive side of work-family interface, not conflict, what you concentrate on, so I don’t think it is the best one for your purposes.

Authors' Response: We did not combine WFC and FWC because the two variables are theoretically and empirically different, as shown in previous studies (Amstad et al., 2011; Mesmer-Magnus & Viswesvaran, 2005; Shockley & Singa, 2011; Whitely & England, 1977). Consistent with existing research on the theoretical difference, we tested WFC and FWC separately in the model. Our findings indicate that WFC and FWC have different mediating effects linking Big Five personality and mental health outcomes. It should also be noted that we cannot combine two meta-analytically derived correlations. Recall that we did not do the meta-analyses ourselves and thus could not recode the primary studies for any that combined both forms of conflict into one construct. We would also need nine more correlations of the combined form of WFC/FWC with each antecedent and outcome in the model. These statistics just do not exist. 

We cited reference number 11 (Byron, 2005) to show that WFC and FWC are different from other work-family constructs such as work-family enrichment. There are positive sides to the work-family interface such as work-family enrichment which is a field still growing and likely to grow more in the future and thus be meta-analyzed and ripe for inclusion in a mega-meta. We would, of course, be happy to remove this citation per your request. 

7. I would ask the same with anxiety and depression, such as have you tried to combine these to general mood or mental health measure and run the analyses with it? Do these combinations affect the effect sizes? Or to the model fit indexes?

Authors' Response: Similar to our response above, these statistics just do not exist in the meta-analytic literature surrounding WFC and FWC. As above, if they existed, we would need a meta-analytic effect size between the combination of depression and anxiety (two disorders that often are comorbid) and every other variable in the model. Thus, if only one or two such meta-analyses existed we would have a very, very simple model as most cells in the 10x10 synthetic input matrix would be empty. 

8. RMSEA shows poor fit, is that a reason to be concerned?

Authors' Response: It is our understanding that the RMSEA is sensitive to model complexity. Our serial mediation model is likely more complex than most. We are pleased that, as we noted in our paper, the SRMR is excellent and the CFI indicates good fit. We hope the excellent SRMR sort of offsets the poor RMSEA, but as with all supplemental fit indices that is a judgment call. If our model had been a simple model with five exogenous predictor variables (the Big Five traits) in the prediction of five endogenous outcome variables (WFC, FWC, depression, anxiety, and substance use) our RMSEA would likely have been better. However, such a simple model belies the extant literature on WFC and to some extent does not give fodder for our use of Conservation of Resources Theory. If all fit indices were poor we would have found some evidence that researchers of the antecedents of WFC and other researchers of the outcomes of WFC might be examining incompatible segments of the nomological network surrounding WFC. We are happy to add more text about the poor RMSEA if the reviewer insists but we hope that the reader will recognize the excellent and good fit of the other indices.

Reviewer 3 Comments: 

1. The authors should pay attention to small errors in the text.

Authors' Response: We apologize for the errors and have made the following corrections: 

Line 47: Replaced the brackets with parentheses around the acronym 

Line 48: Replaced the brackets with parentheses around the acronym 

Line 83: Replaced "[Oh, 2020] with "[23]"

Line 85: Corrected the spelling of "supplemented"

Line 146: Replaced "...pessimism, and temperamentalism" with "...pessimism and being temperamental" 

Line 179: Replaced the brackets around "[substances]" with parentheses

Line 345: Replaced "606" with the correct difference value of "1039"

Line 336: This line appeared twice due to adding text previous to it. We have now renumbered the lines from line 340 onward

Lines 392-394: Replaced brackets around "a", "b", and "c" with parentheses

Lines 491-503: Replaced brackets around trait names (e.g. agreeableness) and hypothesis numbers with parentheses

Lines 564-565: Replaced brackets with parentheses where appropriate

2. Please provide bibliographic references in all parts of the introduction. You cannot affirm something if it is not based on literature.

Authors' Response: In the five paragraphs comprising the introduction section we have 23 sentences. Twelve of the sentences have bibliographic citations. In that section there are 25 citations. The sentences without citations are summary sentences built upon previous sentences with the referenced sources such as line 44-45: "This conflict is likely to have serious consequences for one's roles their family and work." Most of the other sentences without citations are explaining the purpose of the study and how we assembled our input correlation matrix. We are not sure where we might need additional references but would be happy to do so with further guidance. 

3. Please summarize or improve the introduction. It is very long and there are parts that are not understood. It must be more specific and fit the objectives and hypotheses of the study.

Authors' Response: Our introduction section is only five paragraphs long with three of them devoted to the purpose of the study, how "data" were collected/assembled, and the contributions of the study. We respectfully disagree that it should be any shorter than it already is without skipping vital information for the reader. We hoped to show the reader why we wanted to do the research, how it was going to be conducted, and how we hoped to make a contribution to the literature. We are happy to remove some small bits if they can be directly specified. 

4. Regarding the structure of the Introduction section, I would suggest presenting objectives and hypotheses after adequately stating the background and justification rather than in reverse or mixed order for clarity reasons.

Authors' Response: We think that perhaps the reviewer is suggesting changes to the Theoretical Framework section rather than the Introduction section. Our model is rather complicated by its nature as a serial mediation model. We hoped to "walk the reader through" the model by proceeding left to right, that is, from antecedents to mediators to outcomes. Given the very large number of hypotheses we very much prefer to discuss one part of the model and then give its hypotheses then followed by a discussion of the next part of the model and its hypotheses, etc. this sort of piecemeal procedure from left to right also serves our stated purpose of joining the left hand side of the nomological network (see the heading in line 126) with the right hand side (see line 153). We are willing to put all hypotheses at the end of the Theoretical Framework section but they will take up almost three entire pages of text. We think that might be a bit overwhelming to the reader so we hope to stay with the current arrangement. 

5. Please discuss the potential theoretical and clinical implications of your work deeply.

Author's Response: We thank the reviewer for this suggestion but are hesitant to discuss clinical implications. We have lines 517-551 where we discuss the findings within the Theory of Conservation of Resources. For example lines 529-533 reads as follows: "Due to dysfunctional emotional adjustments, sensitivity, and vulnerability to negative experiences [54, 59, 89], neurotic employees have fewer resources with which to manage incompatible work and family demands [46], leading to higher WFC, which in turn increases depression and substance use." However, we have added a section on practical implications to lines 612-627. Our paper's central focus is work-related. The main constructs are WFC and FWC. The personality trait antecedents are measured using instruments designed to measure personality in sub-clinical populations. Admittedly our study does use variables like clinical depression, clinical anxiety, and sub-clinical substance use but it is not a clinical paper, per se. As we state a few times in the paper we used variables that existed, that were found in previous meta-analyses on variables related to WFC/FWC, and could not use variables for which the necessary statistics did not exist. We hope the reviewer understands and we respectfully suggest that such additions to the paper are not in its best interest.

---

## [Editor Report · Decision Letter 1]

24 Jan 2022

Antecedents and Outcomes of Work-Family Conflict:  A Mega-Meta Path Analysis

PONE-D-21-39844R1

Dear Dr. Brian K. Miller,

We’re pleased to inform you that your manuscript has been judged scientifically suitable for publication and will be formally accepted for publication once it meets all outstanding technical requirements.

Kind regards,

Rogis Baker, Ph.D

Academic Editor

PLOS ONE
---

## [Editor Report · Acceptance letter]

28 Jan 2022

PONE-D-21-39844R1 

Antecedents and Outcomes of Work-Family Conflict: A Mega-Meta Path Analysis 

Dear Dr. Miller:

I'm pleased to inform you that your manuscript has been deemed suitable for publication in PLOS ONE. Congratulations! Your manuscript is now with our production department. 

Kind regards, 

on behalf of

Dr. Rogis Baker 

Academic Editor

PLOS ONE